# The Ancient Greek Sophists in Emanuele Tesauro's *Il cannocchiale aristotelico* (1670): Thrasymachus and Gorgias

**Teodoro Katinis**

Department Literary Studies—Italian Literature, Faculty of Arts and Philosophy, Ghent University, 9000 Gent, Belgium; teodoro.katinis@ugent.be

**Abstract:** Emanuele Tesauro's *Il cannocchiale aristotelico* (*The Spyglass of Aristotle*) is widely considered a masterpiece of the Baroque, mainly because of his theory of metaphor as a cognitive tool. But this work is much more than that. Tesauro presents his volume as the ultimate interpretation of Aristotle's rhetorical art, which is clearly not the case. Indeed, his work is a polycentric discourse on a revolutionary theory of rhetoric that goes beyond any previous treatise written on the subject, including Aristotle's *Rhetoric*. Despite his relevance in the history of rhetorical theories, Tesauro's work is still waiting for a comprehensive study of its own as well as investigations of some of its specific aspects. Furthermore, the majority of the existing studies of Tesauro are in Italian (with only very few in English), which makes it difficult for this text to reach an international public. This essay explores what seems to be a specific aspect that has so far been almost completely neglected: the role played by the ancient sophists in the *Cannocchiale aristotelico* and in the history of rhetoric that Tesauro redesigns. Tesauro proclaims his fidelity to Aristotle but actually contradicts Aristotle's anti-sophistic approach. During this analysis, we will discover even more about Tesauro's pro-sophistic attitude: he grounds the climax of Latin rhetorical tradition in Greek sophistry. This positive assessment of the ancient sophists, especially Thrasymachus and Gorgias of Leontini, coexists with a critique of Socrates. Except for Sperone Speroni, no other early modern Italian author—or European author—has proposed this radical inversion of the canon established by Plato. This reversal makes Tesauro a relevant case study in the on-going exploration of the legacy of ancient sophists in Western literature.

**Keywords:** Emanuele Tesauro; sophistic tradition; Seicento

## 1. Introduction

The role of the sophistic tradition in the building of Renaissance culture has recently emerged as a fruitful line of research, especially with regard to the 15th and 16th centuries, when the ancient sophists were revitalized by the humanists' works of translation, and their works were, as a result, widely discussed throughout the Renaissance. NeoLatin and Italian literature was at the center of this process of revitalization, in which the names of Leonardo Bruni, Marsilio Ficino, Pietro Bembo, Erasmus of Rotterdam, Sperone Speroni and Jacopo Mazzoni, among many others, have a central place. The magnitude of the reception of the sophistic tradition and its consequences for the Renaissance is now an established research subject, especially for literary production in Italy and France.[1] Less effort, however, has been devoted to the Italian literature of the 17th century, the so-called Baroque period. Because the Baroque is widely recognized as the climax of rhetorical speculation and activity in Europe, one can advance the general hypothesis that the "*sophistique sacrée*" studied by Marc Fumaroli in French and Italian literature may not be the only form of sophistry in that period.[2] While we await a broader exploration of Baroque Italian literature, this paper limits itself to a single classic in the history of rhetoric: Emanuele Tesauro's *Il cannocchiale aristotelico*.

Tesauro's *Il cannocchiale aristotelico* is a work that enjoyed wide success in Italy and Europe up to the 18th century, with at least 20 editions published in Italy between 1654 (the *editio princeps*) and 1702, with the major part of them having been published in Venice, a cultural hub for the whole of Europe and home to a very efficient printing industry since the beginning of the 16th century. A complete Latin translation of the volume was published in 1698, leading to the wide circulation of this work in Europe.[3] We choose the 1670 edition for our analysis because it follows the last will of the author and presents the latest and most complete version of the text, including a magnificent *anti-porta* that represents the emblematic portal or point of entry to the volume.

Since the rediscovery of Tesauro by Benedetto Croce in the early 20th century, the scholarly production on Tesauro has been mostly in Italian. The paucity of scholarly work on Tesauro in other languages, especially English and French, explains why this Baroque master of rhetorical theory has not garnered the international attention his work deserves. There is no study, for example, of the presence of sophistry in Tesauro's works, with the exception of a few mentions in studies on the *Il cannocchiale*. No English monograph has ever been dedicated to this author, although there are some very fine essays.[4] Even in international anthologies with broad coverage there is no mention of Tesauro, which has also limited international recognition of Tesauro's work on rhetoric. In fact, despite the relevance of Tesauro's *Il cannocchiale* in the history of early modern rhetoric, there exists only one published monograph on this work, though a brilliant monograph by Maicol Cutrì may soon be published.[5]

Thus, when we consider, more specifically, the scholarship on the sophistic tradition in *Il cannocchiale*, we encounter a few pages on the sophistic nature of Tesauro's discourse and no investigation of the presence or use of specific sophists, such as Gorgias of Leontini, in the volume.[6] The aim of this essay, then, is to examine the role of the sophists in Tesauro's *Il cannocchiale*—in particular, Thrasymachus and Gorgias—and assess how the discourse about these sophists is contextualized in the broader purpose of his work, which is to advance the ultimate interpretation of Aristotle's *Rhetoric*. After a brief introduction to some aspects of the *Il cannocchiale*, starting with an enigmatic element in the *anti-porta*, I will provide a close reading of a few passages in which Tesauro uses the ancient Greek sophists to rewrite the history of rhetoric. In the process one will notice the peculiarity of Tesauro's style of argumentation, which was possibly influenced by the "*leggere col rampino*" ("reading with the grapnel") strategy fostered by Giambattista Marino.[7]

## 2. An (Almost) Aristotelian Spyglass

Tesauro's *Il cannocchiale* can be described as an attempt to build a universal semiotic by which all reality is conceived as a system of rhetorical elements produced by God, the first rhetorician, and then, subsequently, by Nature. Man imitates God and Nature's rhetorical activity with his intellect, which is made to produce signs and the relations among them to expand the variety of meanings in any field of human life. This activity is possible thanks to the extraordinary nature of the human intellect and its expressive tools, verbal and non-verbal alike. In this respect, Tesauro walks in the footsteps of the humanist idea of the superior dignity of man. Marsilio Ficino and Pico della Mirandola, to mention two iconic figures of 15th-century humanism, are not explicitly cited in the text, but the whole treatise is clearly inspired by their idea—which goes back to Isocrates and Quintilian—of a superior human dignity that elevates man above other beings by virtue of his participation in the divine. While no specific chapter is focused on this, this human superiority emerges consistently in Tesauro's prose. For example, in *Trattato de' ridicoli*, *Capitolo* XII, which is dedicated to the genre of comedy, Tesauro claims that there is nothing to be ashamed of in discussing low subjects, since the human intellect is like the sun: it sheds light on a lower world without being affected by it. Even more, the intellect is part of the highest nature, God's mind, which created the most divine of all creatures from mud: "Hence you should not be disgusted with philosophizing over filthy matters; to pluck almost from the mud the gems of a noble art: the ray of human intellect being similar to

that of the sun, which has the privilege of always passing among the filthiness while it remains clean. Nay, the human mind partakes of the divine; who with the same divinity dwells in the swamps and in the stars, and of the most sordid lotus fabricated the most divine of corporeal creatures".[8] The theme of man as a miracle and the most admirable of God's creatures is clearly the condition for his ability to create wonderful rhetorical and poetic artifacts. Ficino and Pico never strongly connected this human dignity to a theory and practice of rhetoric, so we can interpret Tesauro's discourse as an original variation *sub specie rhetoricae* of a typical humanistic subject.

Considered in its whole complexity, *Il cannocchiale* aims at building a theoretical framework for a new rhetorical epistemology, which serves the purpose of guiding the reader to produce marvelous and persuasive figures. The *Il cannocchiale* intends to offer, in fact, not only a manual for rhetoricians but also a new gnoseological approach to interpret the entire world, an approach that is very far from Galileo Galilei's metaphor of nature as a book. Indeed, for Tesauro, nature is not exclusively written in mathematical and geometrical signs that only scientists can understand, as Galilei proposes in *Il saggiatore*, but rather in words and images meant to be interpreted figuratively by the human intellect. In this world of signs of any kind, the rhetorician is equipped with the knowledge and skills to build new marvelous and meaningful connections to teach and delight others. If one looks at it from Tesauro's inclusive perspective, Galilei's interpretation of nature might be considered as one of many viewpoints but surely not the most amazing or only true one.

*Il cannocchiale* opens with an *anti-porta* (Figure 1) that, as an *emblema* of the entire book, summarizes in allegorical terms its content and meaning. Two almost identical women are seated across from one another. On the left is the art of poetry (*Poesia*), while on the right we find the art of painting (*Pictura*). *Poesia* is assisted by an old man, Aristotle, who holds up a spyglass before her eyes—a metaphor for his *Rhetoric*—so she can look at the sunspots on the top-right corner of the emblem. The title of the *Il cannocchiale*—*The Spyglass of Aristotle*—is a metaphor and an oxymoron at the same time. The invention of the spyglass, the instrument Galilei pointed towards the sky for the first time and, since then, the symbol of the scientific revolution, is attributed to Aristotle, the figure who (according to the new science) built a false representation of the world that was believed for centuries, and against which Galilei fought. In Tesauro's rhetorical twist, the "*cannocchiale*" becomes "*aristotelico*" and now stands for the ancient art of rhetoric. If Galilei's spyglass allows us to see the sunspots, Aristotle's (metaphorical) spyglass allows us to find the perfections and imperfections of any rhetorical product. The figurative meaning of this instrument is made clear in chapter one when Tesauro declares that he is going to analyze the source of wit (*argutezza*) and the structure and workings of the arts that exploit it by words and images. His guide will be the "divine Aristotle," called in the treatise *the* author (*l'Autore*), implying two meanings: author of the main reference (his *Art of Rhetoric*) and the main *auctoritas* as a reliable source of knowledge. His rhetorical spyglass will reveal all the perfections and imperfections of eloquence: "The divine Aristotle, who every rhetorical secret minutely searched and taught to those who attentively listen to him. Such that we may call his rhetorics a most limpid spyglass to examine all the perfections and imperfections of Eloquence. Speaking therefore of the whole Rhetorical Art, which many denied could be taught to us except by mother nature alone, he said: one can surely find this art, if he considers different compositions, of which, either by chance or by industry, some are good and others evil, and knows with his intellect how to subtly investigate the reasons why these are good and those defective: some provoke nausea and others applause".[9]

But this relationship with Aristotle is more complex than we might think, as we can see already at the very beginning of the volume. I will avoid the temptation to provide an interpretation of the entire *anti-porta* (Figure 1) and instead point out a detail that seems to contradict the principle of verisimilitude one would generally expect in a Baroque work. Looking at the title page, we see the sunspots without the telescope held by *Poesia*. In other words, the reader sees what is *not* supposed to be seen without the spyglass. In



the scholarship on the interpretation of the *anti-porta*, this contradiction has never attracted attention even though it illuminates how the *Il cannocchiale* positions itself towards the reader. My hypothesis is that this detail in the *anti-porta*, right where we are about to enter the book, points to the involvement of the reader in the author's point of view: from there on, we are within the perspective offered by Tesauro's spyglass, which makes it possible for us to see the sunspots and the Aristotelian spyglass pointed toward them at the same time. Tesauro thus offers the reader a bifocal effect: we are already captured by his perspective, already inside the rhetorical machine he built, and we see through his spyglass what poetry sees through Aristotle's spyglass. From now on, as readers, we will participate in Tesauro's rhetorical interpretation of Aristotle's rhetoric and poetics, which brings us to Tesauro's use of Aristotle in service of his own rhetorical theory.

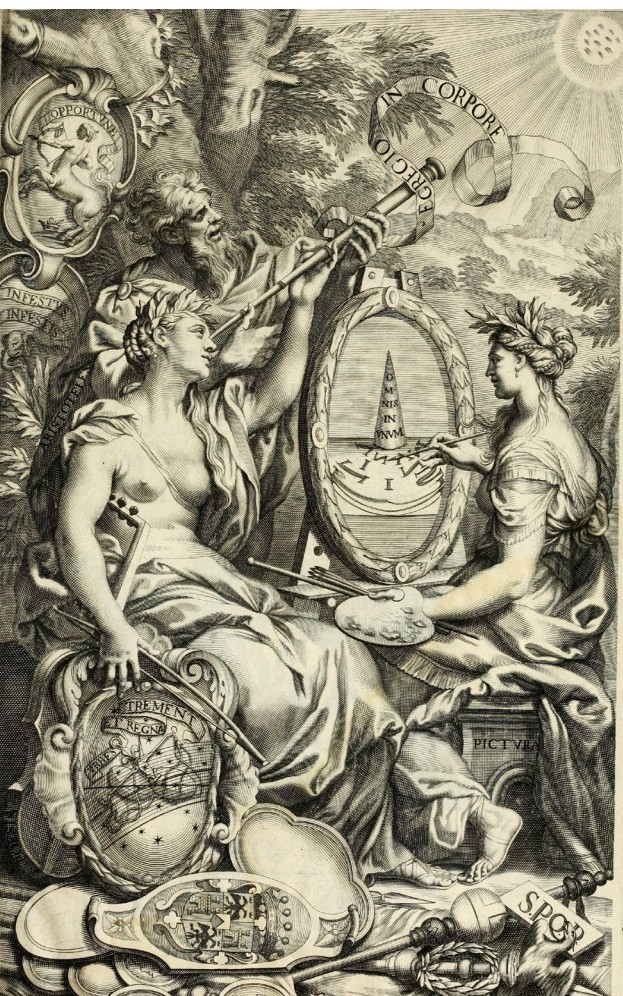

**Figure 1.** Emanuele Tesauro. *Il cannocchiale aristotelico*. Torino, 1670.

Without doubt, Tesauro presents himself as a trustworthy interpreter of Aristotle's works, from which he consistently quotes passages in Latin translation, especially from the *Rhetoric*, in the margins of *Il cannocchiale*. Furthermore, he repeatedly praises Aristotle's authority throughout the volume. Tesauro aims at persuading the reader that his book will unveil the richness of the most precious secrets of Aristotle's rhetorical art for the first time. Ironically enough, this self-presentation is reflected in (Green and Murphy 2006) extremely useful catalogue of Renaissance works on rhetoric, as the *Il cannocchiale* appears among the commentaries on Aristotle, accordingly with the complete title of the work: "The Aristotelian spyglass or idea of the argute and ingenious elocution which serves to the whole oratory, lapidary and symbolic art, examined with the principles of

the divine Aristotle by the Count and Knight of the Grand Cross D. Emanuele Tesauro, Patrician of Turin."[10]

As a matter of fact, Tesauro uses Aristotle to expose his own ideas. A summary of the main elements of his revolutionary rhetoric in *Il cannocchiale* is beyond the scope of this essay, but we can clearly see the distance from Aristotle in the core message of the volume: all levels of reality are involved in a universal semiosis that comprises everything. According to Tesauro, from God's *verbum* and natural phenomena to any arts invented by mankind, the entire world is interpretable in rhetorical and poetical terms. We are exposed here to a literary *trompe l'oeil* effect by which we are invited to see a comment on Aristotle in what is in fact an original work built on a manipulative reading of Aristotle. One of the devices Tesauro uses to support this reading is the sequence of quotes from Aristotle's *Rhetoric* and *Poetics* that he lists at the margins of his text along the whole volume, which are supposed to be evidence for his rigorous method of interpretation.

The reader can certainly appreciate the erudition of the author of *Il cannocchiale*, a former Jesuit who was equipped with deep knowledge of his authors and *auctoritates*, but his use of these authors, including *the* 'Autore' (i.e., Aristotle), is intended as a pretext to craft his own rhetorical theory on an original and erudite interpretation of the ancient philosopher. This is a sophistic operation in itself that could be an intriguing subject for an essay, but my intention here is rather to analyze the mechanism by which an anti-sophistic author—as Aristotle clearly is—can become a pro-sophistic one in Tesauro's text. We will begin by looking at the presence of sophistic rhetoric at the beginning of the *Il cannocchiale* and at the parallel criticism of Socrates.

### 3. Sophistic Excellence and Socrates' Madness

The first time the sophistic tradition is mentioned in the text is in the second subchapter ("*Nome dell'argutezza*") of chapter one, dedicated to the exploration of the origin of the name *argutezza*.[11] Tesauro proposes a mapping of all the translations from Greek to Latin and Italian of the words related to it, from the Aristotelian *schema* and *apophthegma* to the Ciceronian *concinnitas* and *argutia* and the Italian *motto arguto* and *concetto arguto*, in a vortex of names that, according to Tesauro, are connected to each other across authors and centuries. A climax in this rhapsodic history of ideas and terms surrounding the *argutezza* is the era of sophists, who brought rhetoric to its artistic pinnacle with their epideictic eloquence. One passage focuses on the excellence of their style, which won the favour of the judges, which was also praised by Cicero: "And which style was sharper and more ingenious than that of the Sophists, and Declamators, who composing only by ostentation of sharp wit, made of every clause an argument, of every argument a concept, and with their concepts obtained from the judges the victory: *Nihil est* (says Tullius) *quod illi non assequantur suis Argutijs*. They came finally with the same name after Persius, Quintilian and Aulus Gellius, who telling us that Favorinus praised the fever, added: *Expergificando ingenio, vel exercendis Argutijs*."[12]

After the first generation of sophists, a second generation of Greco-Roman sophists emerged, who performed their eloquence during the so-called Second Sophistic. At the beginning of this passage in *Il cannocchiale*, the sophists are praised for being able to convince the judge "*per ostentation d'ingegno*" (for, but also by, the pleasure of showing their skillfulness). At the end of the passage, epideictic (or demonstrative) rhetoric is highlighted by the example of Favorinus' oration in praise of fever. Tesauro knew very well that among the three genres of rhetoric treated in Aristotle's *Rhetoric*, namely, deliberative, judicial, and epideictic, the third—typical of the sophistic tradition evoked in Tesauro's passage—is the least esteemed genre and widely viewed as the "Cinderella" of the three *genera*.[13] The rhetorical turn of the Renaissance, fueled by the rediscovery of the ancient sophists in the 15th and 16th centuries, elevated the epideictic genre above the other two, as is evident, for example, in Sperone Speroni's *Dialogo della retorica* (1542), in which epideictic rhetoric is compared to a sun that projects its warm, lively rays upon deliberative and judicial rhetoric and thus as the eye of the entire rhetorical art.[14] This was already a

significant deviation from Aristotle's text before the Baroque age. In this respect, Tesauro inherits a special attention to the demonstrative genre from the previous century and emphasizes its function in creating a sense of the marvelous. Going back to the quoted passage from *Il cannocchiale*, the most interesting part is Tesauro's discussion of the sophists' ability to make a "concept" from every argument. "*Concetto*" is a classical key term in Baroque literary culture, while the "*concettismo*" was the new poetics initiated in Italy at the end of the 16th century.[15] This reconnecting of the sophistic tradition to a fulcrum of Baroque poetics is an important contribution by Tesauro, who brings the ancient sophists into his work as precursors of the "*concettismo*" even as he projects, retrospectively, this same "*concettismo*" onto ancient Greece.

We find the counterpart to this praise of the sophists at the beginning of chapter two when Tesauro defines the *argutia archetipica*, or archetypical image. That is what we picture in our imagination before depicting it in actual forms and colours to be communicated to our interlocutor. Because of our human nature, we cannot communicate directly from our imagination to the interlocutor, which creates the need to practice the rhetorical arts to delight our audience. Following anecdotes reported by several sources, including Cicero, Vitruvius, and Lucian, Tesauro blames Socrates, who wished that human beings had a small window on their chest to exchange concepts ("*concetti*") without linguistic means. According to Tesauro's reading of the sources, Socrates thought that language confuses human communication and betrays our intentions. But Nature, Tesauro argues, could compose an apologetic speech saying that if Socrates' wish came true, she would deny human beings of all the delight coming from the performance of eloquence.  In the personification of Nature as an orator defending herself against the twisted wish of Socrates, Tesauro suggests an image of a distorted man who went against Nature. Since Plato, Socrates has usually been considered the example of a philosopher who struggled against the sophistic tradition of his time. But there is at least one early modern precursor who objected to this interpretation of the sophists and attacked Socrates for his warped nature and way of thinking. This is, again, Speroni, who wrote two "*trattatelli*", *In difesa dei sofisti* and *Contra Socrate*, to argue, against Socrates and Plato, for the fundamental role of sophists in the ancient republics.[16] What is interesting here is the similar position in the *Il cannocchiale*: Tesauro praises the sophists' eloquence while attacking Socrates' philosophy, although Tesauro's purpose is not to emphasize the importance of sophistic rhetoric for political life but rather to reveal the function of eloquence as a means to produce wonder and learning. It is worth citing the entire passage from *Il cannocchiale* within which Socrates' foolishness is contextualized: "Archetypal wit is that which we paint in our souls by thinking, as if imagining I say to myself: I take a porcupine hurling its arrows around me, to threaten my enemies, both near and far. And this archetypal wit is that whose effect we intend to color in the soul of others by outward symbols, not being permitted to pass it on from spirit to spirit, without the ministry of the senses. And this was the foolish anger of Socrates, blaming nature for not having opened a little window in the chest of men to see face to face the original of their concepts, without interpretation of lying language; whose translations are often betrayals. Against which complaint, nature could compose her apologetic; answering that she would at one and the same time defraud the ingenious of the delight of so many beautiful arts of speech."[17]

## 4. The Sophistic Roots of Latin Eloquence

The brief but relevant mention of the sophistic tradition at the beginning of *Il cannocchiale* is not an isolated case in the volume. On the contrary, Tesauro expands on this when he analyzes in more detail the characteristics of sophistic eloquence and its place in the history of ancient rhetoric. Because of the importance of the passage involved in Tesauro's argumentation in defense of sophists, I present the entire text in translation in an Appendix A at the end of this essay.

This text is in the section titled *Harmonic figures* (*FIGURE HARMONICHE*). Furthermore, this section is the major part of chapter four, titled *Formal cause of the wit. On the*

*figures (CAGION FORMALE DELL'ARGUTIA. Circa le figure)*, a quite long chapter (Tesauro 2000, pp. 121–206) that examines different types of rhetorical figures that can be considered witty (not all of the figures, of course, can be considered so). Tesauro claims that this examination requires the retracing of the true genealogy of rhetorical figures. After a long journey through numerous definitions given by ancient authors, among whom Aristotle is considered to be the most reliable source, Tesauro concludes that rhetorical figures are means by which one transforms a normal instance of communication into an appealing oration by making it an enjoyable learning of a novelty, so that the listener can learn by enjoying and enjoy by learning: "I conclude that the rhetorical figures are nothing other than a pilgrim quirk, varying the oration from the everyday and vulgar style so that it has teaching combined with novelty, and the listener at the same time learns by enjoying and enjoys by learning".[18] Furthermore, human beings are satisfied at the levels of sensibility, affection, and intelligence, and we therefore have three different genres of figure (*HARMONICO, PATETICO, ET INGEGNOSO*),[19] each appealing to one of these three forms of satisfaction. The "harmonious" figures are meant to appeal to the hearing with the harmonic resonance of the sentence. The rest of the chapter is therefore dedicated to this genre of figure. Tesauro points out that for a long time in ancient Greece orators used a "pending" oration (ORATIONE PENDENTE) characterized by long, wordy, uniform, and monotonous prose with no attention to the pleasing of the ears, which had already been criticized by Aristotle. In this type of speech there was only a period at the end of the entire piece, with no pause and breathing during the process, as if it were but the sound of cicadas.[20] This style influenced the Latin orator as well as the first Italian vernacular writers, including Giovanni Boccaccio and some historians.[21] But not all Greek orators followed the mainstream. Two great orators, Thrasymachus and Gorgias, diverged and initiated a new style that forever changed the history of rhetoric as a practice. And here begins the part that is most interesting to us and that can be read in the Appendix A.

I provide here a brief summary of Tesauro's argumentation, though I leave the reader the pleasure of having a direct contact with Tesauro's typical rhetorical style in writing about rhetoric, an effect that deserves to be enjoyed without a filter (except the translation, which might be necessary). Tesauro introduces Thrasymachus as the first who perceived the unpleasantness of the "pending" orations and looked to poetry to find a solution. Hence, he broke the long orations down into short sentences with intervals and called them "periods" (PERIODI), which, syntactically, is a relatively independent unit. These periods could finally bring renewed sweetness to the listeners' ears, although the ancients did not know the cause of this effect. After Thrasymachus, a revolution was brought by Gorgias along the same path, who broke those periods into even shorter pieces and created new periods, or clauses, characterized by their concinnity. This gave a poetic rhythm to the prose, so that the speech became even more delightful and marvelous to listeners. Gorgias' style enters Roman eloquence through Cicero, who adopted it in the last years of his career. This Gorgianic style of Cicero became the model for later Latin authors and, in fact, in some of them it seemed that Gorgias himself was reborn ("*ne' quali parve rinato Gorgia Leontino*").

Is this narration historically reliable? It certainly poses some issues related to the way in which Tesauro treats his sources and manipulates them to achieve his goals. It is not the intention of this essay to attempt an exploration of these sources, but only to present the case of Aristotle, the main authority of *Il cannocchiale*, because this is the author that Tesauro quotes explicitly and with reference to the specific passages from his works—most of them being taken from the *Rhetoric*—that are put in the margins of the *Il cannocchiale* to direct the reader to the Aristotelian source. By doing so, we will not point to any exploration of the largest spectrum of Tesauro's sources. Indeed, such an exploration would bring us into a very broad area of readings and possibly the labyrinth of an author with a Jesuit education who bends all his sources to serve a specific purpose. In either case, we are not interested here in reconstructing the ideal library of Tesauro but in showing how the very author (Aristotle) that Tesauro presents as the highest authority on

rhetoric can legitimize the rehabilitation of two ancient sophists (Thrasymachus and Gorgias), who are consequently viewed as the source of the highest figure of the Latin rhetorical tradition, namely, Cicero. Neither Aristotle nor Cicero would have agreed with this description of the history of rhetoric, and Tesauro was certainly aware of this fact.

The very passages from Aristotle that Tesauro places in the margins of his text serve the purpose of *Il cannocchiale* only if taken out of their context. Indeed, once we read them in the context of the argumentation of Aristotle's *Rhetoric* the manipulative operation performed by Tesauro reveals itself. But the reader is silently invited to believe in the coherency between Tesauro's text and the paratextual quotes, and in the existence of a secret message in Aristotle's words. The reshaping of Aristotle's intentions includes, among other things, supporting sophistry instead of condemning it.

With this context in mind, we will now analyze the three quotes used by Tesauro in his discourse on the harmonic figures (reported also in the passage in our Appendix A). The first passage mentioned by Tesauro is from *Rhetoric* 3.9, in which Aristotle treats the structure of the sentence as a part of the broad topic of style. More specifically, in this chapter Aristotle analyzes the difference between paratactic and hypotactic prose in terms of effects on the audience. Tesauro quotes a Latin translation (*Periodum supinam appello que uno membro constat*) in which Aristotle defines the term that Tesauro translates in Italian as '*periodo*'. I offer here the English translation of the passage from which Tesauro extracts one sentence (the one italicized). After a previous passage in which Aristotle defined the "extended style," he continues: "This then is the extended style, but the contracted style is that in periods. *By period I mean a clause having a beginning and end in itself* and an easily surveyed magnitude. This is both pleasant and easily learned […]" (1409a).[22]

Tesauro then praises this style made of short clauses because of the delight it conveys to listeners' ears, and to support his argument he quotes another passage from the *Rhetoric* (3.8) in which Aristotle treats prose rhythm as a part of the broader subject of style. Tesauro quotes the Latin translation (*Restat etiam Pean: quo quasi secreto a Trasimaco invento, incipientes utebantur: sed nesciebant dicere quis esset*) of the passage, in which Aristotle treats the paean genre in the margins of his text. The original context from which this passage (here italicized) is drawn says*: "There remains the paean, which speakers have used from the time of Thrasymachus, but without being able to say which it was.* […] The other rhythms, then, are to be rejected for the reasons given and because they are metric; but the paean should be adopted; for it alone of the rhythms mentioned is not a metre, so that it is the most easily concealed" (1409a).[23]

The third and last Latin passage (*Ex membris constare dicitur, cum oratio absoluta sit, et distincta; facilisque respirationis. Membrum vero est altera particula*) quoted by Tesauro (and here italicized) in relation to the "harmonic figures" comes from the same chapter two of the first quote: "The *period* is either in clauses or straightforward. *Now a clausal style is that which is complete and partitioned and can be delivered without drawing breath, not in division but as a whole* (the clause being one or other of its parts). **By a straightforward period I mean that with a single clause**. Now, both the clauses and the periods should neither taper away nor be lengthy" (1409b).[24] In the same passage of chapter nine we also find a second definition (the passage highlighted in bold) that could be an additional source of inspiration for Tesauro's definition of "period," to be put beside the first of the three quotes listed above.

Now, if we go back to the full argumentation of Tesauro, reported in our Appendix A, we notice that he is giving credit to Thrasymachus that is not at all found in Aristotle text, in which the sophist is simply mentioned to say that at his time the paeanic style was adopted for rhetorical practice. As a matter of fact, Tesauro uses a few of Aristotle's sentences to support a broader narrative that praises the *periodo* and the sophist Thrasymachus as its founder. And this is in the broader context of *Il cannocchiale*, in which Tesauro presents himself as the ultimate interpreter of the true message of Aristotle's rhetorical work. Furthermore, Tesauro's praise of Thrasymachus is put at the beginning of an historical excursus that sees Gorgias not only as a step forward in the evolution of rhetoric

but as its climax: Gorgias is a model for the following orators, among whom the best ones are presented as virtual pupils of Gorgias in whom Gorgias himself is reborn.

## 5. Conclusions

If we consider the scope of Emanuele Tesauro's masterwork, *Il cannocchiale Aristotelico*, there are only a few passages in which the ancient sophists are mentioned. But most of these passages, including the ones treated in this contribution, praise the sophistic tradition as a source of important revolutionary ingredients in the history of rhetoric. This approach is difficult to harmonize with the presentation of the *Il cannocchiale* as a trustworthy commentary on Aristotle's philosophy, and in particular his rhetorical theory. As a matter of fact, in Aristotle's text on rhetoric there is no praise of the sophists as a category of rhetoricians, while his discussion of the sophists Thrasymachus and Gorgias—the ones highlighted by Tesauro in his chapter as sources of the highest Latin eloquence—is always related to specific questions about their style and offers no praise for them as sophists.

Extrapolating Aristotle's quotes from their original context, connecting them to a new text as marginal paratexts, and integrating them in the new narrative built throughout *Il cannocchiale* are meta-rhetorical strategies, before and above the specific ones within the text, that Tesauro designed to persuade the reader that his revolutionary rhetorical theory was already discussed by Aristotle, the ultimate ancient authority on rhetoric. Moreover, this theory also includes a rehabilitation of the ancient sophists as models and historical milestones for the rhetorical art *in toto*. This is how the Aristotelian spyglass reveals itself to be, in fact, Tesauro's spyglass. And it is Tesauro's spyglass that allows readers to see the sunspots in the *anti-porta* or emblem that serves as the emblematic exordium to *Il cannocchiale*. The perfections and imperfections of rhetoric—and the perception of Aristotle's philosophy as well—are a product of Tesauro's discourse, in which the ancient sophists and their legacy play a major role, both historically and theoretically, as a turning point in equipping the art of eloquence with fundamental resources for persuasion.

**Funding:** This research received no external funding.

**Conflicts of Interest:** The authors declare no conflict of interest.

## Appendix A. Transcription and Translation of a Passage from *Il cannocchiale*

*Appendix A.1. Criteria of the Transcription and Translation*

In my transcription and translation of *Il cannocchiale* I modernize the text for the following aspects: punctuation and use of diacritical marks; use of "u" and "v" as well as "s" and "f" have been distinguished. But I have kept the use of capital letters because they are elements of Tesauro's style that do not pose any obstacle to the understanding of the text for a modern reader. I also reproduce in the footnotes the Latin citations from Aristotle's *Rhetoric* that Tesauro quotes in the margins of his text in the 1670 editions to support his discourse. Tesauro's original text is followed by my English translation. Because of the unique features of Tesauro's prose and vocabulary, I have opted for a translation that aims mainly to make the text accessible, which therefore may diminish the variety and polysemy of his prose but will at least make it understandable for a non-specialist or non-native speaker.

*Appendix A.2. Transcription of (Tesauro 2000, pp. 126–27)*

*FIGURE ARMONICHE. […] Trasimaco adunque fu il primo ingegno che, osservando la Pendente Oratione ascoltarsi con altretanta spiacenza con quanto diletto le Liriche Odi erano udite, avvisò seco stesso questa differenza procedere dalla grata vicenda delle pause et dalla soavità delle poetiche misure. Cominciò egli pertanto a minuzzar la massa di quelle anaboliche Clausulone in brievi'ntervalli chiamandoli PERIODI, cioè Rivolgimenti, ad esempio et misura delle Strofe et Antistrofe che partivano le Odi Pindariche, sovente respirando et rivolgendosi a capo. Hor queste sue periodi Ritonde et, come le nomina il nostro Autore,[25] Supine, quantunque con un sol tratto di*

*penna sonoramente corressero et, nascondendo sovente aguisa delle serpi la testa nella coda, serbassero il verbo in fine, trovò egli nondimeno per virtù della prosodia una segreta modulatione nel principio, nel corso et nella fine ch'empieva gli orecchi di nuova et maravigliosa dolcezza.*[26] *Talché gli Uditori, conoscendo l'effetto dell'Arte senza conoscere l'Arte, godevano ad udirlo et non sapevano la cagione. Ma come facil cosa è l'aggiugnere agli trovati altrui, Gorgia Leontino, più diligente osservatore, fabricò di queste Periodi ritonde Periodi concise*[27] *trinciandole in piccole clausulette, chiamate Membra et Articoli gratiosamente corrispondenti et misurati tra loro. Onde la Periodo supina e piana, divenendo misurata e concisa, non più ritonda, né però mozza, non metrica né senza metro, non ligata né sciolta dalle poetiche leggi, senza verso, non senza ritmo, parendo verso a' prosatori et prosa a' versificatori, era agli uni et agli altri maravigliosamente gradita. Entrò in pregio fra' Romani questa pellegrina merce (com'io ti narrai) negli ultimi anni di Cicerone, il cui stilo tanto si arrotò su quella cote forense et sì divenne acuto che possiam dire haver percosso Verre di piatto et Antonio di punta. Quinci egli stesso confessò la beltà di queste Periodi concise, in odio delle ritonde già sue familiari et favorite, dicendo: Iucundior est Periodus, si est articulis membrisque distincta; quam si continuata et producta: quia suas respirationes habet: et mens respirat cum Oratore: Deinde magis dilucida est, qui memoria facilius tenetur; et magis patet. Le quali cosiderationi buonamente copiò dal nostro Autore. Nè senza molta argutezza queste Periodi figurate et concise chiamò egli CONCINNITATES. Onde di Gorgia disse: Cuius in Oratione numerum plerumque efficit ipsa Concinnitas. Et allo incontro dello stile di Eschilo et Eschine: In ijs erat admirabilis cursus Orationis (ecco la periodo ritonda) Ornata sententiarum Concinnitas non erat. Quasi e' paragoni la Periodo Supina alle belle chiome, ma sparte et cadenti, et la Concisa et figurata alle medesime chiome divise in ciocche, ciascuna delle quali, vibrata col caldo calamistro, s'increspa et inanella. Della qual metafora facetamente si serviva Augusto, chiamando le Retoriche figure del suo favorito, CINCINNOS MECOENATIS. Tanto è, che da que' tempi la Romana eloquenza, deposto il Manto et le cadenti maniche di quello stile Asiatico et ritondo, incominciò camminare alla Spartana, succinta in Attica vesticella, et in iscambio di scettro vibrò lo strale. O fosse genio delle attempate orecchie di Augusto, divenute implacabili nemiche delle parole, o novello studio delle solinghe Academie di que' nobili Declamatori, Cestio, Asinio, Argentario, Seneca, Portio Ladrone, Arellio, Silone et Osco; a' quali feteva ogni periodo non acuminata et concisa. Et per le lor vestigie caminarono dapoi Plinio Cecilio, Nazario, Ausonio, e tutti que' famosi Panegiristi ne' quali parve rinato Gorgia Leontino.*

*Appendix A.3. English Translation*

*Harmonic figures.* […] Thrasymachus was the first genius that, paying attention to the unpleasant feeling in listening to the hanging oration and the delight of listening to the lyric odes, understood that this difference comes from the use of the pauses and the suavity of the poetic measures. Therefore, he started to break down the mass of anabolic clauses in order to obtain short intervals that he called periods, or cycles. For instance, the measure of the stanzas and anti-stanzas with which the Pindaric Odes begun, often by breathing and going to the next step. Now, about these rounded periods of his, or *supine*, as our Author calls them, although they sonorously run by a sole trait of pen, they hid the head in the tail like a snake and they hold the verb at the end, he found in them nonetheless, by the virtue of prosody, a secret modulation at the beginning, during the period and at the end, which fill the ears with a new and marvelous sweetness. Therefore, the audience, which knew the effect of his art without knowing the art, enjoyed listening to him without knowing the reason why. Being easy to add new things to the other's discoveries, Gorgias of Leontini, who was a more diligent observer, crafted from those rounded periods concise periods by cutting them into small clauses, called limbs and articles, which graciously correspond to and are measured upon each other. In consequence this supine and plan period was pleasant for anyone, since it was figurative and concise, not rounded anymore but not cut off, not metric but not without any meter, not bound to either, independent from poetical rules, without verse but not without rhythm, and it appeared as poetry to prose writers and as prose to poets. This foreign merchandise was appreciated by the Romans (as I narrated) in the last years of Cicero, whose style was so much shaped

by the forensic activity and became so acute that we could say that he stroked Verres by cut and Antonius by thrust. And then he himself declared that he liked these concise periods and hated the rounded ones with which he was familiar and used to like, by saying: "*iucundior est periodus, si est articulis membrisque distincta; quam si continuata et producta: quia suas respirationes habet: et mens respirat cum oratore: deinde magis dilucida est, quia memoria facilius tenetur; et magis patet*", which are observations that he took from our Author. And not without much wit, Cicero called concinnities these figurative and concise periods, and he therefore said about Gorgias: "*cuius in oratione numerum plerumque efficit ipsa concinnitas*" and, on the contrary, on the style of Aeschylus and Aeschines, "*in ijs erat admirabilis cursus orationis* (here is the rounded period) *ornata sententiarum concinnitas non erat*," as if the supine period were beautiful but scattered like hanging manes, whereas the concise and figurative period was the same manes but divided into locks, each one rippled and curled with a warm curling iron. Augustus wittily used this metaphor by calling "*cincinnos mecoenatis*" the rhetorical figures of his favorite. Since then, the Roman eloquence deposed the mantle and the falling sleeves of that Asiatic and rounded style, started walking in a Spartan manner dressed with a succinct Attic clothing, and exchanged her scepter with a dart to throw. This happened because of the understanding of Augustus' old ears which became implacable enemies of words, or because of the new method of study of the solitary academies of the noble declaimers Cestius, Asinius, Argentarius, Seneca, Porcius Latro, Arellius, Silo, and Oscus, for whom any not sharpened and not concise period stank. And then their footprints were followed by Plinius Caecilius, Nazarius, Ausonius, and all the other well-known panegyrists in whom it seemed that Gorgias of Leontini was reborn.

**Notes**

1.　　See (MacPhail 2011; Katinis 2018, 2019) for the main results.
2.　　See (Fumaroli 1980) for the "*sophistique sacrée*".
3.　　For a history of the text, see (Cutrì 2020).
4.　　The most recent studies are (Snyder 2016, 2017).
5.　　The most recent and complete study of the *Il cannocchiale* is (Cutrì 2021), a very fine book (with an up-to-date bibliography) we hope to see published soon. The first and only published monograph on the *Il cannocchiale* is (Frare 2001). For a presentation of Tesauro and his work, see also (Merola 2007) and (Bisi 2019).
6.　　"*il carattere sofistico dell'acutezza*" is mentioned in (Bisi 2011, p. 35), while a more extended discussion is in (Tedesco 2003, pp. 33–70). The subject would deserve a further analysis, but this is beyond the scope of this contribution.
7.　　See (Cutrì 2021, p. 71) for Marino's text.
8.　　"*Hor non dei tu havere a schifo il filosofar sopra Materie schifose; per coglier quasi dal fango le gemme di un'Arte nobile: essendo il raggio dell'humano Intelletto simile a quel del sole, che ha privilegio di trascorrere sempre mondo fra le immondezze. Anzi la mente humana partecipa della Divina; che con la medesima Divinità habita nelle paludi, et nelle stelle: et del più sordido loto, fabricò la più Divina delle Corporee Creature*" (Tesauro 2000, p. 584).
9.　　"*il Divino Aristotele, che ogni Rettorico secreto minutamente cercò e tutti gli 'nsegnò a color che attenti l'ascoltano. Talché possiam chiamar le sue Rettoriche Un limpidissimo CANNOCCHIALE per esaminar tutte le perfettioni et le imperfettioni della Eloquenza. Parlando egli dunque di tutta l'Arte Rettorica, la qual molti pur negavano potercisi 'nsegnare, se non dalla sola Madre Natura, disse: colui sicuramente poterne ritrovar l'Arte, il qual propostosi Componimenti diversi, de' quali, o per caso, o per industria, sian' altri buoni e altri mali, sappia col suo ingegno sottilmente investigar le ragioni, perché questi siano ottimi et quegli difettosi: gli uni movan nausea e gli altri applauso.*" (Tesauro 2000, pp. 2–3).
10.　"*Il CANNOCCHIALE ARISTOTELICO O sia Idea DELL'ARGUTA ET INGEGNOSA ELOCUTIONE Che serve a tutta l'Arte ORATORIA, LAPIDARIA, ET SIMBOLICA Esaminata co' Principij DEL DIVINO ARISTOTELE Dal Conte et Cavalier Gran Croce D. EMANUELE TESAURO PATRITIO TORINESE*". See (Green and Murphy 2006, p. 41) (in the section *Commentaries* on Aristotle) and 429–430 (for a list of editions).
11.　(Tesauro 2000, pp. 4–9).
12.　"*Et quale Stilo fu più acuto et ingegnoso di quel de' Sofisti, e Declamatori, che componendo solo per ostentation di acuto ingegno facean di ogni Clausula un Argomento, di ogni Argomento un Concetto, et co' suoi Concetti ottenean da' Giudici la Vittoria: Nihil est (dice Tullio) quod illi non assequantur suis Argutijs. Vennero finalmente col medesimo Nome appresso a Persio, Quintiliano et Aulo Gellio, il qual dicendoci che Favorino laudò la Febre, soggiunse: Expergificando ingenio, vel exercendis Argutijs*" (Tesauro 2000, p. 8).
13.　(Vickers 1988, p. 54).
14.　See (Katinis 2018, chapter 2) for Speroni's text and the analysis of it.

15. See, for example, Camillo Pellegrino, *Del concetto poetico*, written in 1598 and published in (Borzelli 1898), in which a young Giovan Battista Marino discusses with his interlocutors the identity and function of a "*concetto*" in poetry.

16. See (Katinis 2018) for the "*trattatelli*" in original and English translation and *passim* for the analysis of their topics.

17. "*ARGUTIA ARCHETIPA è quella che noi ci dipingiamo nell'animo col Pensiero, come se imaginando io dico intra me: 'Io prendo un'Histrice scagliante gli suoi strali dognintorno, per minacciare a' miei nimici, così vicini, come lontani'. Et questa Argutia Archetipa è quella il cui protratto intendiamo di colorir nell'animo altrui per via de' simboli esteriori, non essendoci permesso di tramandarlo da spirito a spirito, senza il ministero de' sensi. Et questa fu la sciocca rabbia di Socrate, incolpante la Natura del non havere aperto una finestretta in petto agli huomini, per veder faccia a faccia l'Originale de' lor concetti, senza interpretamento di lingua mentitrice; le cui traditioni sovente son tradimenti. Contro alla qual querela potea compor la Natura il suo apologetico; rispondendo ch'ella harebbe ad un tempo defraudato gli 'ngegnosi del diletto di tante belle Arti sermonali*" (Tesauro 2000, p. 16).

18. "*Conchiudo, le Figure Rettoriche altro non essere che Un vezzo pellegrino, variante la Oratione dallo stile cotidiano et vulgare accioch'ell'habbia insegnamento congiunto con la novità, et l'uditore in un tempo impari godendo et goda imparan*" (Tesauro 2000, p. 124).

19. *Ibid*.

20. *Ibid*.

21. (Tesauro 2000, p. 125).

22. (Aristotle 1991, p. 232).

23. (Aristotle 1991, p. 231).

24. (Aristotle 1991, p. 232).

25. Ar. 3 *Rhet.* C. 9: *Periodum supinam appello que uno membro constat.*

26. Ar. 3 *Rhet.* C. 8: *Restat etiam Pean: quo quasi secreto a Trasimaco invento, incipientes utebantur: sed nesciebant dicere quis esset.*

27. Ar. 3 *Rhet.* C. 9: *Ex membris constare dicitur, cum oratio absoluta sit, et distincta; facilisque respirationis. Membrum vero est alteraparticula.*

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
