# Peer review of "The Ancient Greek Sophists in Emanuele Tesauro’s Il cannocchiale aristotelico (1670): Thrasymachus and Gorgias"

_humanities, doi:10.3390/h13010033_

Round 1

Reviewer 1 Report

Comments and Suggestions for Authors

Overall view:

This article is worth publishing because it makes an important point, as mentioned below. There are some matters that need to be attended to, but I do not consider them overall to be major issues.

Abstract:

There are basically two types of abstracts: the first explains what an article is going to discuss; the second provides a summary of what the article achieves. I much prefer the second type of abstract. The author essentially combines these two types by explaining what is going to be discussed in the article and only toward the end of the abstract explains what the article achieves. The abstract could be condensed, with a focus on the last one-third to last one-half of its content. (I do not know what the word limit is for abstracts for the journal Humanities, but it seems to me that the abstract exceeds the length of what a journal normally would accept for an abstract.)

Argument:

One of my areas of interest and knowledge is Roman rhetoric. Although I have an interest in Greek rhetoric and the Renaissance, I do not consider myself an expert on the former and I am not familiar with Emanuele Tesauro except for having heard of his name. But the article seems to me to make an important point about Tesauro's positive view of the ancient Greek sophists as being important sources of Latin eloquence, an argument that the author claims has hitherto been neglected by scholars. I am prepared to accept this statement of the author on face value. There is nothing startling about the major thesis of the article, but the author still makes an important point, in my view, in the short article.

Endnote numbers:

Why do the endnote numbers appear in Roman numerals instead of Arabic numerals (lines 511-535)?

Citation:

Lines 42-46: It would be good to provide a list or at least a reference to where there is a list. It is not clear if f.n. iii contains this list or only applies to the Latin translation.

Line 85: where in Iscorates and Quintilian?: the ancient references should be cited when making a claim such as this.

Texts and Translations:

Lines 413-509: It is good to have provided the Italian text and English translation of a passage from Tesauro's Il cannocchiale.

The method of citing ancient sources needs to be improved. The system of citation is not scholarly. The ancient work, for example, of Aristotle, should be cited (see, e.g., n. 1), but it should be the original Greek reference that appears, not merely a page number to a Penguin English translation. The English translation can be cited as an addendum to the citation of the original Greek reference and scholarly edition of the Greek text. One can do this even if one is unfamiliar with ancient Greek. The same would apply to the citation of any Latin translation.

Other Comments:

Lines 107, etc. It is good to have written "Galileo Galilei" the first time and then "Galilei" afterward, as you have done.

Lines 525 and 526: Something is missing. What does "X" mean? I am missing something here. Cf. n. 1: "MacPhail 2011; X and X." Again, I am not understanding something here. Or are these omissions that are to be included later?

Comments on the Quality of English Language

English Style:

The English is all right, but I am not a fan of the first-person style, which I find somewhat self-indulgent and diffuse. It is much better simply to make assertions than to write, "I will avoid", "we will provide", "we will notice" (who exactly is "we"?), etc., which basically just takes up space, though I am aware that this academic style has become popular. Sometimes the personal pronouns are mixed incongruously in the space of two sentences (e.g., "we might think", I will avoid", lines 144-145).

As I have indicated, the English is generally all right, but there are some awkward sentences. For example, the sentence between lines 400 and 404 is awkwardly long and could be rewritten or divided into two sentences.

Author Response

Thank you for your useful comments.

Reviewer 2 Report

Comments and Suggestions for Authors

This is a very interesting essay that will be of great interest to people working on rhetoric and on Italian literature.

Comments on the Quality of English Language

There are just a few details to clean up. I corrected the manuscript using track changes.

Author Response

Thank you for your useful comments. 

Reviewer 3 Report

Comments and Suggestions for Authors

Given the paucity of scholarship on Tesauro it could be worth mentioning in the opening footnote the 1975 edition of his treatise on imprese by M.L. Doglio. Also, it could be mentioned the chapter by Giovanni Baffetti in La predicazione del Seicento, also ed. by M.L. Doglio (and C. Delcorno).

P. 3, l. 147: I am not entirely sure that verisimilitude is what one would "generally" expect from a Baroque work. Maybe reword slightly?

In the transcription, the Latin quotes in the Italian text could be put in roman font (they are in italics in the English translation). Also, I would add a space between "brievi" and "''ntervalli" and separate "aguisa"

Author Response

Thank you for your useful comments. 

Reviewer 4 Report

Comments and Suggestions for Authors

Author Response

Thank you for your useful comments. 

Round 2

Reviewer 2 Report

Comments and Suggestions for Authors

everything looks good